# DW2009 Elevates the Efficacy of Donepezil against Cognitive Impairment in Mice

**DOI:** 10.3390/nu13093273

**Published:** 2021-09-19

**Authors:** Dong-Yun Lee, Jeon-Kyung Kim, Soo-Won Yun, Myung Joo Han, Dong-Hyun Kim

**Affiliations:** 1Neurobiota Research Center, Department of Life and Nanopharmaceutical Sciences, College of Pharmacy, 26 Kyungheedae-ro, Seoul 02447, Korea; dongyun8246@naver.com (D.-Y.L.); kim_jk0225@naver.com (J.-K.K.); 2Department of Food and Nutrition, Kyung Hee University, 26 Kyungheedae-ro, Dongdaemun-gu, Seoul 02447, Korea; ysw6923@naver.com (S.-W.Y.); mjhan@khu.ac.kr (M.J.H.)

**Keywords:** cognitive impairment, lipopolysaccharide, donepezil, *Lactobacillus plantarum* C29, DW2009

## Abstract

*Lactobacillus plantarum* C29 and DW2009 (C29-fermented soybean) alleviate cognitive impairment through the modulation of the microbiota-gut-brain axis. Therefore, we examined whether combining donepezil, a well-known acetylcholinesterase inhibitor, with C29 or DW2009 could synergistically alleviate cognitive impairment in mice. Oral administration of donepezil combined with or without C29 (DC) or DW2009 (DD) alleviated lipopolysaccharide (LPS)-induced cognitive impairment-like behaviors more strongly than treatment with each one alone. Their treatments significantly suppressed the NF-κB^+^/Iba1^+^ (activated microglia) population, NF-κB activation, and tumor necrosis factor-α and interleukin-1β expression in the hippocampus, while the brain-derived neurotropic factor (BDNF)^+^/NeuN^+^ cell population and BDNF expression increased. Their treatments strongly suppressed LPS-induced colitis. Moreover, they increased the Firmicutes population and decreased the Cyanobacteria population in gut microbiota. Of these, DD most strongly alleviated cognitive impairment, followed by DC. In conclusion, DW2009 may synergistically or additively increase the effect of donepezil against cognitive impairment and colitis by regulating NF-κB-mediated BDNF expression.

## 1. Introduction

Dementia is a disorder characterized by a progressive decline in cognitive function. Alzheimer’s disease (AD) is the most common neurodegenerative disease showing dementia [1]. The pathophysiology of AD is still unclear, except for the genetic inheritance [2]. Many hypotheses, including the abnormal deposit of the amyloid β protein, aggregate of hyperphosphorylated tau protein, cholinergic neuron damage, inflammation, and oxidative stress, have been suggested [2,3,4]. Based on these hypotheses, anti-AD drugs have been developed [2]. For instance, cholinesterase inhibitors and cholinergic agonists are frequently used for the therapy of AD [5,6]. Of these, donepezil (Aricept), a well-known acetylcholinesterase inhibitor, reversibly inactivates acetylcholinesterase and thus inhibits acetylcholine hydrolysis, resulting in increased acetylcholine concentrations at cholinergic synapses. Recently, to increase the therapeutic efficacy of donepezil, a variety of combined prescriptions, such as donepezil/memantine (an antagonist for N-methyl-D-aspartate receptor), donepezil/AC-3933 (a partial inverse agonist for newly developed benzodiazepine receptor), and donepezil/choline alphoscerate (a choline-containing phospholipid) have been developed [6,7,8]. These combined prescriptions synergistically or additively attenuate cognitive impairment.

Probiotics are live bacteria and yeasts with potential health benefits [9]. Of these microorganisms, many Lactobacilli and Bifidobacteria have been developed as functional foods. They alleviate gut dysbiosis [10,11], suppress colitis [12], modulate the immune response [13], suppress body weight gain [14], and improve psychiatric disorders, such as depression and cognitive decline [15,16]. *Lactobacillus helveticus* NS8 attenuates cognitive decline in chronic stress-induced mice [17]. *Lactobacillus mucosae* NK41 suppresses *Escherichia coli*-induced memory impairment with the attenuation of gut dysbiosis, suppression of hippocampal NF-κB activation, and induction of hippocampal brain-derived neurotrophic factor (BDNF) expression in mice [18]. *Bifidobacterium longum* NK46 improves cognitive decline in 5XFAD mice by alleviating gut dysbiosis and reducing bacterial lipopolysaccharide (LPS) production [16]. *Lactobacillus plantarum* C29 attenuates LPS- or 2,4,6-trinitrobenzenesulfonic acid (TNBS)-induced memory impairment in mice [19,20]. The C29 supplementation (DW2009, C29-fermented soybean) alleviates cognitive decline with high blood BDNF levels in individuals with mild cognitive decline via the modulation of the microbiota-gut-brain axis [21]. However, whether probiotics including C29 or their supplementations can elevate the efficacy of donepezil against cognitive impairment remains elusive.

Therefore, we investigated whether combining donepezil with *Lactobacillus plantarum* C29 or DW2009 could synergistically alleviate LPS-induced cognitive impairment in mice. 

## 2. Materials and Methods

### 2.1. Materials

Donepezil and LPS were purchased from Sigma (St Louis, MO, USA). DW2009 (C29-fermented soybean flour) was donated from DongHwa Pharm Institute (Gyunggi-do, Korea).

### 2.2. Culture of Lactobacillus Plantarum C29

*Lactobacillus plantarum* C29 (KCCM11291P), used in the preparation of DW2009, was supplied from the Korea Culture Center of Microorganisms. C29 is a gram-positive and non-hemolytic bacterium. C29 was cultured in commercial media for probiotics, including De Man, Rogosa, and Sharpe (MRS) broth (BD, Franklin Lakes, NJ, USA) at 37 °C for 10 h. Cultured bacteria were centrifuged and freeze-dried. DW2009 was prepared by defatted soybean flour fermentation with *Lactobacillus plantarum* C29 and in combination with *Lactobacillus plantarum* C29 [21]. 

### 2.3. Animals

C57BL/6 mice (male, 6 weeks old, 19–22 g) were fed with water and food ad libitum, and maintained under the controlled condition (light cycle, 12 h [07:00–19:00]; temperature, 22 °C ± 1 °C; humidity, 50% ± 10% humidity). Mice were acclimatized for 7 days before the usage of experiments. All experiments were approved by the Committee for the Care and Use of Laboratory Animals at Kyung Hee University (IACC, KHUASP(SE)-20176) and were carried out according to the Kyung Hee University Guidelines for Laboratory Animals Care and Use.

### 2.4. Preparation of Mice with Cognitive Impairment

Mice with cognitive impairment were developed by the intraperitoneal injection of LPS (10 μg/kg/day), as previously reported [22]. Each group consisted of seven mice. 

First, the efficacies of C29 and DW2009 were evaluated. Mice were divided into four groups (NC, LP, C29, and DW). LPS was injected intraperitoneally in mice, except for the normal control group (NC), once a day for 5 days. NC received saline instead of LPS. Test agents (NC, vehicle [saline] alone; LPS, vehicle; C29, 1 × 10^9^ CFU/mouse/day of C29; and DW, 100 mg/kg/day of DW2009) were orally administered once a day for 5 days from 24 h after the final treatment with LPS.

Second, to understand whether C29 and DW2009 could enhance the efficacy of donepezil, the effects of their combinations were examined in mice with LPS-induced cognitive impairment. Mice were divided into seven groups (NC, LP, C29, DW, DP, DC, and DD). LPS was injected intraperitoneally in mice, except for NC, once a day for 5 days. NC received saline instead of LPS. Test agents (NC, vehicle [saline] alone; LPS, vehicle; DP, 1.5 mg/kg/day of donepezil; C29, 1 × 10^9^ CFU/mouse/day of C29; DW, 100 mg/kg/day of DW2009; DC, 1.5 × 10^9^ CFU/mouse/day of C29 plus 1.5 mg/kg/day of donepezil; and DD, 100 mg/kg/day of DW2009 plus 1.5 mg/kg/day of donepezil) were orally administered once a day for 5 days from 24 h after the final treatment with LPS. The dosage of donepezil was calculated from the FDA draft guideline. 

Cognitive function was evaluated in the Y-maze, novel object recognition (NOR), and Barnes maze tasks 18 h after the final treatment with test agents. The test agents were administered once a day during the performance of behavioral tasks. The mice were sacrificed 6 h after the final behavioral task. For the immunoblotting and ELISA assay, the hippocampus and colon were removed and stored at −80 °C until use. For the immunofluorescence assay, mice were perfused transcardiacally with 4% paraformaldehyde.

### 2.5. Behavioral Tasks

The Y-maze, NOR, and Barnes tasks were carried out according to the method of Lee et al. [22]. Each mouse was initially placed in the center of the Y-maze, and within one arm, a three-arm horizontal maze (40 [length] × 3 [width] × 12 cm [height]) and the sequence of arm entries were recorded for 8 min. A spontaneous alternation was defined as an entry into all three arms consecutively. The spontaneous alteration (%) was indicated as the ratio of the spontaneous to possible alternations. The NOR task was carried out in a black rectangular open field apparatus (45 × 45 × 45 cm). In the first trial, each mouse was explored in the apparatus in the presence of two identical objects, and the number of times that each object was touched was recorded for 10 min. The second trial was performed 24 h after the first trial; each mouse was explored in the apparatus in the presence of one old object used in the first trial and one new object. The exploration (%) was indicated as the ratio of the number times touching the new object to the number times touching all objects. The Barnes maze task was carried out in an apparatus consisting of a circular platform (89 cm, diameter) with 20 holes (5 cm, diameter) located evenly around the perimeter and an escape hole box. The training finished after each mouse entered the escape hole or after the maximum test duration (5 min), and each mouse was allowed to stay in the escape hole for 30 s. If the mouse failed to get into the escape box within 5 min, it was led to the escape hole. Mice were given two trials each day for 5 consecutive days. The latency time to reach the escape hole was recorded.

### 2.6. Myeloperoxidase Activity Assay, Immunblotting, and Enzyme-Linked Immunosorbent Assay (ELISA)

The brain and colon tissues were lysed in radio immunoprecipitation assay lysis buffer (Pierce, Rockford, IL, USA) and centrifuged (10,000× *g*, 4 °C, 10 min). The resulting supernatants were used for myeloperoxidase activity assay, immunoblotting, and ELISA assay. The assay of myeloperoxidase activity was performed according to the method of Jang et al. [22]. BDNF, p65, p-p65, CREB, p-CREB, and β-action expression levels were assayed by immunoblotting according to the method of Jang et al. [22]. IL-1β, IL-10, and TNF-α expression levels were assayed using commercial ELISA kits (Ebioscience, Atlanta, GA, USA) [22].

### 2.7. Immunofluorescence Assay

Brain and colon tissues were removed from perfused mice, post-fixed with paraformaldehyde solution (4%), cytoprotected in sucrose solution (30%), and freezed. The frozen tissue was sectioned [22]. The section was washed with phosphate-buffered saline, blocked with serum, incubated with antibodies for NF-κB (1:100, Cell Signaling, Danvers, MA, USA), Iba1 (1:200, Abcam, Cambridge, UK), CD11c (1:200, Abcam), BDNF (1:200, Millipore, Burlington, MA, USA), and/or NeuN (1:200, Millipore) for 16 h, and incubated with the secondary antibodies conjugated with Alexa Fluor 594 (1:200, Invitrogen, Carbsband, CA, USA) or Alexa Fluor 488 (1:200, Invitrogen) for 2 h to visualize. Nuclei were stained with 4′,6-diamidino-2-phenylindole dilactate (Sigma, St Louis, MO, USA). The immunostained section was scanned with a confocal laser microscope and the entire intensities of the single plane images were quantified by ImageJ software (github.com/imagej/imagej1).

### 2.8. Microbiota Composition Analysis

Genomic DNA was purified from the feces of mice using a QIAamp DNA stool mini kit (Qiagen, Hilden, Germany) and amplified using barcoded primers for the V4 region of the bacterial 16S rRNA gene according to the method of Kim et al. [18]. Each amplicon was sequenced using Illumina iSeq 100 (San Diego, CA, USA). The sequenced reads were deposited in the NCBI’s short read archive (accession number, PRJNA739073).

### 2.9. Statistical Analysis

Experimental results are described as mean ± SD using GraphPad Prism 8 (GraphPad Software, Inc., San Diego, CA, USA). Significant differences were analyzed using one-way ANOVA followed by Duncan’s multiple range test (*p* < 0.05). All F and *p* values are indicated in Appendix A.

## 3. Results

### 3.1. Effects of C29 and DW2009 in the LPS-Induced Cognitive Impairment in Mice

In the preliminary study, the effects of C29 and DW2009 on LPS-induced cognitive impairment was examined in mice (Figure 1). The intraperitoneal injection of LPS significantly caused cognitive impairment: it reduced spontaneous alteration in the Y-maze task and exploration in the NOR task to 74.2% and 73.4% of NC, respectively. However, oral administration of C29 and DW2009 significantly increased LPS-suppressed spontaneous alteration in the Y-maze task to 95.8% and 96.9% of NC, respectively (Figure 1B). However, the means of the arm entry numbers in the Y-maze task were not significantly different between all tested groups (Figure 1C), which demonstrated that general locomotor activity was not affected by C29 or DW2009 treatment, as previously reported [23]. Oral administration of C29 or DW2009 significantly recovered LPS-suppressed exploration in the NOR task to 96.9% and 98.8% of NC, respectively (Figure 1D).

### 3.2. C29 and DW2009 Increased the Efficacy of Donepezil on LPS-Induced Cognitive Impairment in Mice

To examine whether C29 and DW2009 could enhance the efficacy of donepezil, the combined effects of donepezil with and without C29 or DW2009 on LPS-induced cognitive impairment were investigated in mice (Figure 2A). Exposure to LPS increased cognitive impairment-like behaviors: it reduced spontaneous alternation in the Y-maze task and exploration in the NOR task, as well as increased latency time in the Barnes maze task compared to that of NC (Figure 2B–D). However, oral administration of donepezil, C29, or DW2009 significantly reduced LPS-induced cognitive impairment-like behaviors in the Y-maze, NOR, and Barnes maze tasks. Furthermore, when donepezil was combined with C29 or DW2009, the cognitive impairment-ameliorating activities of these combinations were more potent than each one alone. Of these, the combination of donepezil with DW2009 (DD) alleviated LPS-suppressed cognitive impairment-like behaviors the most strongly, followed by DC and DW2009 ≈ donepezil ≈ C29. 

Donepezil, C29, and DW2009 significantly suppressed LPS-induced activation of NF-κB and expression of IL-1β and TNF-α in the hippocampus (Figure 2E–G). They reduced the population of NF-κB^+^/Iba1^+^ cells while the population of BDNF^+^/NeuN^+^ cells was increased (Figure 2H). They also induced the expression of IL-10, claudin-5, and BDNF and phosphorylation of CREB, while NF-κB activation was suppressed by DW2009 alone (Figure 2I). DD and DC suppressed LPS-induced the activation of NF-κB, expression of IL-1β and TNF-α, and population of NF-κB^+^/Iba1^+^ cells (activated microglia) in the hippocampus more strongly than donepezil, while the population of BDNF^+^/NeuN^+^ cells, expression of BDNF and IL-10, and phosphorylation of CREB increased (Figure 2E–I). 

### 3.3. DC and DD Alleviated Colitis in Mice with LPS-Induced Cognitive Impairment

Exposure of mice to LPS significantly caused colon shortening, increased myeloperoxidase activity, and induced TNF-α and IL-1β expression in the colon, while IL-10 expression was decreased (Figure 3A–E). Consequentially, LPS treatment caused colitis. However, treatment with C29, DW2009, or donepezil reduced LPS-induced colon shortening, myeloperoxidase activity, and TNF-α and IL-1β expression and increased IL-10 expression in the colon, leading to in the alleviation of colitis. Of these, DW2009 alleviated colitis most potently. Furthermore, DD and DC more potently decreased myeloperoxidase activity and TNF-α, IL-1β, and IL-6 expression than DW2009 and C29, respectively. They also significantly reduced NF-κB activation, NF-κB^+^/CD11c^+^ cell population, and claudin-1 expression (Figure 3F,G).

### 3.4. DC and DD Modulated the Gut Microbiota Composition in Mice with LPS-Induced Cognitive Impairment

Gastrointestinal inflammation causes gut dysbiosis. Intraperitoneal injection of LPS caused gastrointestinal inflammation, as previously reported [22]. Therefore, we investigated whether the alleviation of LPS-induced cognitive impairment by oral administration of donepezil, DC, or DD was associated with gut microbiota (Figure 4, Appendix A). Exposure to LPS shifted the bacterial β-diversity (principal coordinate analysis [PCoA]), while the bacterial α-diversity (OTUs) was not affected. However, treatment with C29, DW2009, donepezil, DC, or DD partially shifted the bacterial β-diversity of LPS-treated mice to that of NC, while the α-diversity was not affected. At the phylum level, DW, DC, and DD significantly decreased the Bacteroidetes population, while the Firmicutes population increased. Furthermore, they increased the Lachnosipracease and Lactobacillaceae populations at the family level, Lactobacillus, PAC000664_g at the genus level, and EF097112_s, *Lactobacillus murinus* group, and KE159538_g_uc populations at the species level. However, they decreased the Prevotellaceae population at the family level, Prevotella and Prevotellaceae_uc population at the genus level, and EU622763_s group, PAC001070_s group, and PAC002445_s populations at the species level.

Next, we analyzed the correlation between gut microbiota composition at the family level and cognitive function or proinflammatory cytokines (Figure 5). Sphingobacteriales_uc, Bacteroidales_uc, and Prevotellaceae populations were negatively correlated with spontaneous alternation in the Y-maze task, while the Bacteroidaceae population was positively correlated with it (Figure 5A, Appendix A). Enterococcaceae and Bacteroidaceae populations were positively correlated with exploration in the NOR task, while the Bacteroidales_uc and Lachnospiraceae populations were negatively correlated with it. The Bacteroidales_uc population was negatively correlated with the latency time in the Barnes maze task, while Bacteroidaceae and Rikenellaceae populations were positively correlated with it. Furthermore, Bacteroidales_uc and Prevotellaceae populations were positively correlated with TNF-α expression level in the hippocampus, while Bacteroidaceae and Citrobacteriaceae populations were negatively correlated with it (Figure 5B). The Bacteroidales_uc population was also positively correlated with IL-1β expression levels in the hippocampus, while Bacteroidaceae and Rhodospirillaceae populations were positively correlated with it. 

## 4. Discussion

Gut microbiota communicate to the brain through nervous, endocrine, and immune pathway [24,25]. Gut microbes produce a variety of byproducts including endotoxins, neurotransmitters, hormones, and short-chain fatty acids, such as LPS, acetylcholine, dopamine, serotonin, adrenaline, and butyric acid [25,26]. The overproduction of gut bacterial LPS by stressors such as TNBS, immobilization, and antibiotics cause a leaky gut through gastrointestinal inflammation, leading to psychiatric disorders such as AD and depression with systemic inflammation via the increase in the blood LPS level [15,27,28]. For instance, exposure to *Escherichia coli* causes cognitive impairment with gut inflammation in mice through an increase in the blood LPS level [27]. A single peritoneal injection of LPS activates astrocytes in the brain and repeated injections of LPS activate microglia, leading to the impairment of neural circuit functions and cognitive function [29]. The intraperitoneal injection of TNBS-induced gut bacterial LPS causes cognitive impairment in mice [27]. The intracerebroventricular administration of LPS causes cognitive impairment through the suppression of NF-κB-mediated expression of BDNF and phosphorylation of CREB [30]. The LPS from gut bacteria *Bacteroides fragilis* activates the inflammatory neurodegeneration-involved NF-κB signaling pathway in human primary brain cells [31]. These findings suggest that excessive, chronic exposure to LPS can impair cognitive function through microglia activation. Anti-inflammatory agents, such as resveratrol, protect cognitive impairment. These results suggest that substances that can inhibit LPS-induced inflammation may be useful for AD therapy. 

In the present study, the repeated injection of LPS induced activation of NF-κB and expression of IL-1β and TNF-α and suppressed expression of BDNF and phosphorylation of CREB in mouse brain, resulting in cognitive impairment, as previously reported [22]. Donepezil, C29, and DW2009 significantly alleviated LPS-induced cognitive impairment-like behaviors in mice in the Y-maze, NOR, and Barnes maze tasks and neuroinflammation in the hippocampus. They also alleviated colitis. Furthermore, they reduced LPS-induced NF-κB^+^/Ib1^+^ and LPS^+^/Iba1^+^ cell populations, IL-1β and TNF-α expression, and NF-κB activation in the hippocampus and NF-κB^+^/CD11c^+^ cell and NF-κB activation in the colon, while BDNF^+^/NeuN^+^ cell population, IL-10 and BDNF expression, and CREB phosphorylation increased. They also alleviated LPS-induced gut microbiota alteration. The efficacy of DW2009 or C29 was comparable to that of donepezil. 

Donepezil is known to alleviate dementia by inhibiting acetylcholinesterase activity [5]. Nevertheless, to increase the efficacy of donepezil, a well-known acetylcholinesterase inhibitor for dementia therapy, it should be used in combination with NMDA receptor antagonists, such as memantine; benzodiazepine receptor partial inverse agonists, such as AC-3933; or a choline-containing phospholipid choline alphoscerate [6,7,8]. Probiotics such as C29 and its supplementation DW2009 strongly inhibit LPS-induced activation of NF-κB and induce the expression of BDNF in vitro and in vivo [20]. Nevertheless, studies on the combined effects of donepezil with probiotics for dementia therapy remain elusive. 

In the present study, C29 and DW2009 additively increased the efficacy of donepezil on LPS-induced cognitive impairment-like behaviors, neuroinflammation, and colitis in mice. DC and DD, which were the combination of donepezil with C29 and DW2009, respectively, increased the population of BDNF^+^/NeuN^+^ cells, expression of BDNF, and phosphorylation of CREB in the hippocampus and decreased the activation of NF-κB and population of NF-κB+/Ib1+ cells in the hippocampus and activation of NF-κB and population of NF-κB^+^/CD11c^+^ cells in the colon. DC and DD alleviated LPS-induced cognitive impairment-like behaviors in the Barnes maze task more strongly than DW2009 or C29 alone, while BDNF expression and CREB phosphorylation in the hippocampus were more strongly increased. They also more strongly suppressed NF-κB activation and NF-κB^+^/CD11c^+^ cells in the colon. In particular, DD more strongly suppressed NF-kB activation and increased BDNF expression than DC. BDNF increases the synaptic plasticity of neuron cells through the induction of glutamatergic synaptic transmission and long-term potentiation [22,23,27]. The BDNF expression level is lower in the hippocampus of patients with AD compared to that in healthy individuals [23]. These findings suggest that C29 and DW2009 can enhance the efficacy of donepezil by inducing NF-κB-mediated expression of BDNF. 

Gut microbiota display important roles in the onset of neuropsychiatric disorders via the bidirectional gut-brain axis [25]. Gut microbiota LPS activates the inflammatory NF-κB signaling pathway, leading to colitis and cognitive impairment [2,20]. We found that intraperitoneal injection of LPS caused gut microbiota alteration, colitis, neuroinflammation, and cognitive impairment-like behaviors. Lee et al. reported that the endotoxin level was higher in 5XFAD and aged mice than in young mice [27]. These results suggest that LPS exposure can cause gut inflammation and neuroinflammation, resulting in the occurrence of gut dysbiosis and cognitive impairment. In addition, antibiotics-induced gut dysbiosis causes gut inflammation, which provokes cognitive impairment through the neuroinflammation. C29, DW2009, and/or donepezil modulated the gut microbiota alternation with cognitive impairment. In particular, DC and DD significantly decreased the population of Bacteroidetes including Prevotellaceae and Cyanobacteria, and increased the population of Firmicutes including Lachnosipracease and Lactobacillaceae. Interestingly, they increased LPS-suppressed Bacteridaceae to Prevotellaceae ratio and decreased the Bacteroidales_uc to Bacteroidaceae ratio. LPS treatment increased neuroinflammation and colitis with the induction of TNF-α and IL-1β expression in the colon and hippocampus. TNF-α and IL-1β expression were positively correlated with cognitive impairment-like behaviors and Prevotellaceae and Bacteroidales populations in the gut microbiota. However, LPS treatment suppressed IL-10 expression and the Bacteroidaceae population, which were negatively correlated with cognitive impairment-like behaviors. These results suggest that gut dysbiosis, such as the induction of Prevotellaceae and Bacteroidales populations and suppression of Bacteroidaceae population can cause cognitive impairment with colitis and C29 and DW2009 may alleviate cognitive impairment with neuroinflammation by modulating gut microbiota composition.

## 5. Conclusions

Oral administration of C29 or DW2009 can alleviate cognitive impairment and colitis. Furthermore, DW2009 or C29 can be gut microbiota-friendly substances and can synergistically increase the efficacy of donepezil against cognitive impairment and colitis. DC and DD may alleviate cognitive impairment by the regulation of NF-κB-mediated BDNF expression through the attenuation of gut dysbiosis and inflammation.

## Figures and Tables

**Figure 1 nutrients-13-03273-f001:**
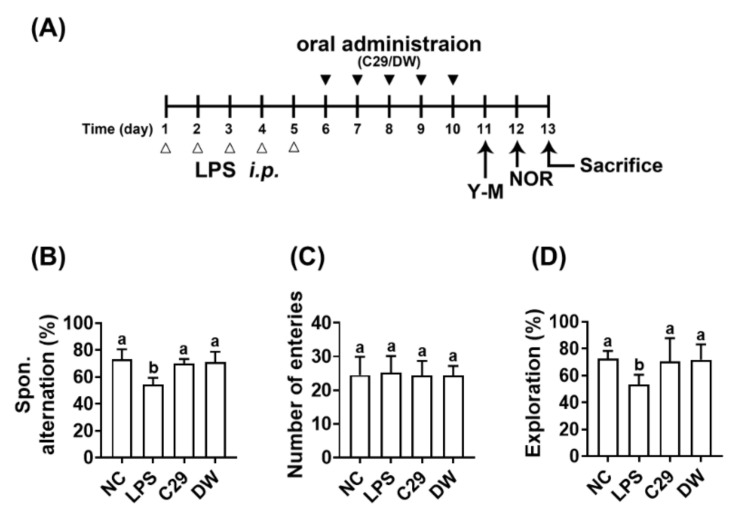
C29 and DW2009 alleviated LPS-induced cognitive impairment in mice. (**A**) Experimental schedule. (**B**) Effects on spontaneous alternation in the Y-maze task. (**C**) Effects on the number of the arm entries in the Y-maze task. (**D**) Effects on exploration in the novel object recognition task. NC was treated with saline instead of LPS (white triange). Test agents (black triangle: LPS, vehicle alone; C29, 1 × 10^9^ CFU/mouse/day of C29; DW, 100 mg/kg/day of DW2009) were orally gavaged in mice daily for 5 days after the intraperitoneal injection of LPS. Thereafter, behavioral tasks in the Y-maze (Y-M) and novel object recognition apparatus (NOR) were performed. Data values are as mean ± SD (*n* = 7). Means with same letters are not significantly different (*p* < 0.05).

**Figure 2 nutrients-13-03273-f002:**
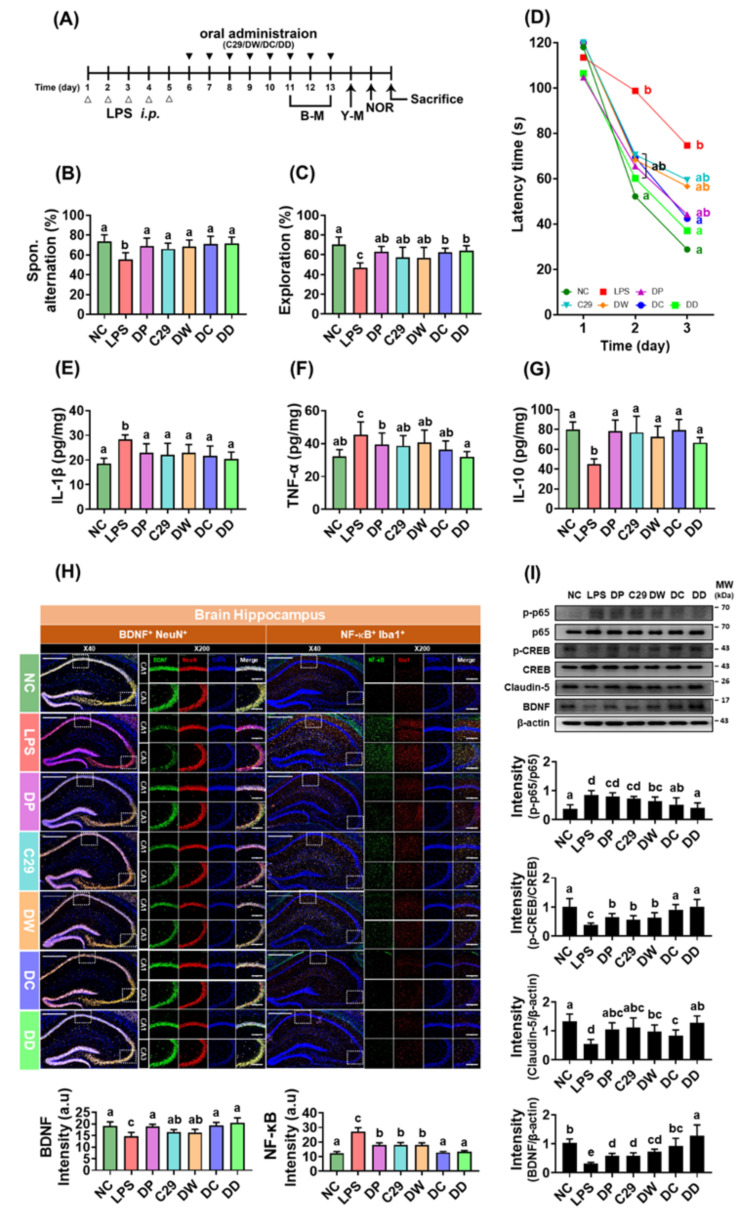
C29 and DW2009 elevated the efficacy of donepezil in mice with LPS-induced cognitive impairment. (**A**) Experimental schedule. (**B**) Effects on spontaneous alternation. (**C**) Effects on exploration. (**D**) Effects on the latency time. Effects on hippocampal IL-1β (**E**), TNF-α (**F**), and IL-10 expression (**G**), assessed by ELISA. (**H**) Effects on BDNF^+^/NeuN^+^ and NF-κB^+^/Iba1^+^ cell populations (**H**) and NF-κB activation, CREB phosphorylation, and BDNF and claudin-5 expression (**I**) in the hippocampus. LPS was injected intraperitoneally in mice daily for 5 days. NC was treated with vehicle (saline) instead of LPS (white triangle). Test agents (black triangle: NC, vehicle [saline] alone; LPS, vehicle; DP, 1.5 mg/kg/day of donepezil; C29, 1 × 10^9^ CFU/mouse/day of C29; DW, 100 mg/kg/day of DW2009; DC, 1.5 × 10^9^ CFU/mouse/day of C29 plus 1.5 mg/kg/day of donepezil; and DD, 100 mg/kg/day of DW2009 plus 1.5 mg/kg/day of donepezil) were orally gavaged in mice daily for 5 days. Thereafter, behavioral tasks in the Y-maze (Y-M) and novel object recognition apparatus (NOR) were performed. The confocal microscopy images of a single plane (H) were quantified by ImageJ software. Magnification scale bars: ×40, 500 μm; ×200, 10 μm. Data values are as mean ± SD (*n* = 7). Means with the same letters are not significantly different (*p* < 0.05).

**Figure 3 nutrients-13-03273-f003:**
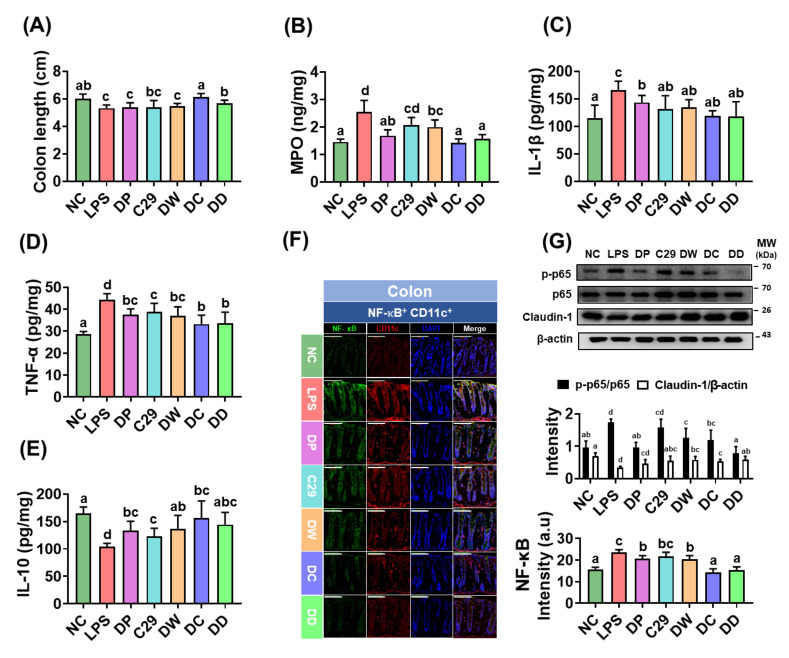
The combined treatment of donepezil with C29 or DW2009 alleviated colitis in mice with LPS-induced cognitive impairment. Effects on colon length (**A**); myeloperoxidase (MPO) activity (**B**), and IL-1β (**C**); TNF-α (**D**), and IL-10 expression (**E**) in the colon. Effects on NF-κB+/CD11c+ cell population (**F**) and NF-κB activation and claudin-1 expression (**G**). LPS was intraperitoneally injected in mice once a day for 5 days. NC was treated with saline instead of LPS. Test agents (NC, vehicle [saline] alone; LPS, vehicle; DP, 1.5 mg/kg/day of donepezil; C29, 1 × 10^9^ CFU/mouse/day of C29; DW, 100 mg/kg/day of DW2009; DC, 1.5 × 10^9^ CFU/mouse/day of C29 plus 1.5 mg/kg/day of donepezil; and DD, 100 mg/kg/day of DW2009 plus 1.5 mg/kg/day of donepezil) were orally gavaged in mice daily for 5 days. The confocal microscopy images of a single plane (**F**) were quantified by ImageJ software. Magnification scale bars are 100 μm. Data values are as mean ± SD (*n* = 7). Means with the same letters are not significantly different (*p* < 0.05).

**Figure 4 nutrients-13-03273-f004:**
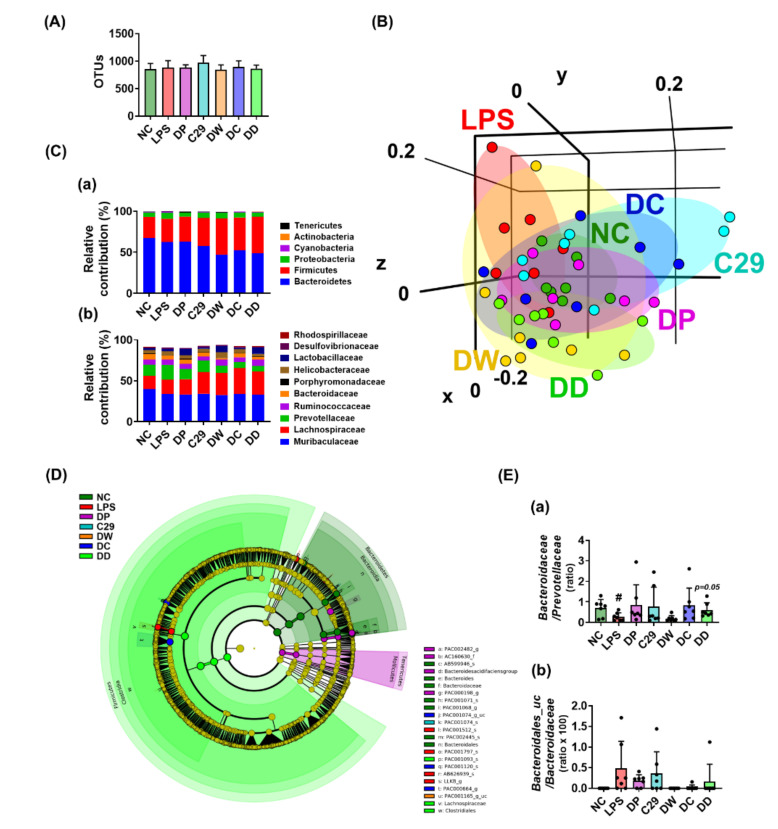
C29, DW2009, donepezil, and their mixtures modulated gut microbiota composition in mice with LPS–induced cognitive impairment. (**A**) Effects on α-diversity (OTUs). (**B**) Effects on β-diversity (principal coordinate analysis (PCoA) plot based on Bray Curtis). (**C**) Effect on the composition of gut microbiota: (**a**) at the phylum level and (**b**) at the family level. (**D**) Effects on the gut microbiota composition, described by Cladogram. (**E**) Effects on the ratio of Bacteroidaceae to Prevotellaceae population (**a**) and Bacteroidales_uc to Bacteroidaceae population (**b**). LPS was intraperitoneally injected in mice daily for 5 days. NC was treated with saline instead of LPS. Test agents (NC, vehicle [saline] alone; LPS, vehicle; DP, 1.5 mg/kg/day of donepezil; C29, 1 × 10^9^ CFU/mouse/day of C29; DW, 100 mg/kg/day of DW2009; DC, 1.5 × 10^9^ CFU/mouse/day of C29 plus 1.5 mg/kg/day of donepezil; and DD, 100 mg/kg/day of DW2009 plus 1.5 mg/kg/day of donepezil) were orally gavaged in mice once a day for 5 days. The behaviors were measured 24 h after the final treatment with test agents. Data values are as mean ± SD (*n* = 7). ^#^ *p* < 0.05 vs. NC group.

**Figure 5 nutrients-13-03273-f005:**
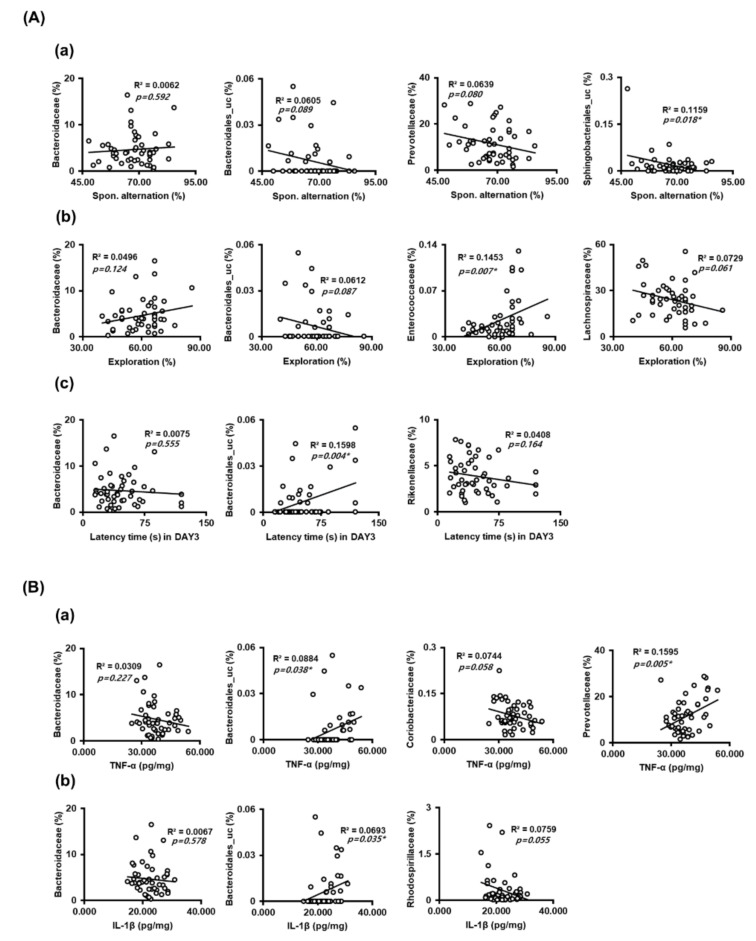
The correlation between the gut microbiota populations and cognitive function or hippocampal proinflammatory cytokines. (**A**) The correlation between the gut microbiota populations and cognitive function in the Y-maze (**a**), novel object recognition apparatus (**b**), and Barnes maze (**c**). (**B**) The correlation between the gut microbiota populations and proinflammatory cytoknes TNF-α (**a**) and IL-1β (**b**).

## Data Availability

The datasets used and/or analyzed during the current study are available from the corresponding author on reasonable request.

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
