# Peer review of "DW2009 Elevates the Efficacy of Donepezil against Cognitive Impairment in Mice"

_nutrients, 2021, doi:10.3390/nu13093273_

Round 1
Reviewer 1 Report
The study of the microbiome-gut-brain relationship is in need of more exploration. Lee and colleagues explored this relationship using lipopolysaccharide (LPS)-induced cognitive impairments with donepezil and various combinations of probiotics. This study has high translational implications across multiple fields. In the manuscript titled “DW2009 elevates the effect of donepezil against cognitive impairment in mice,” the authors investigate the cognitive ability in mice following LPS-exposure along with the Alzheimer’s therapy of donepezil in combination with Lactobacillus plantarum C29 and DW2009 (a C29-fermented soybean).
To improve the manuscript for publication, below are a few items for consideration:
- Throughout the manuscript, check for consistent and accurate grammar and spelling. For example, on line 50, Lactobacillus helveticus should be in italics, and on line 96, the ‘was’ should be a ‘were’, and on line 104, the first sentence is not a complete sentence. There were several other locations in the manuscript with other errors that impact the readability of the manuscript.
- The initial sentences of the introduction are scattered and could be connected better. Aging is discussed here, but not brought up again in the manuscript. This could be removed and the connection to inflammation and NF-kB signaling connected in the next sentence.
- The mice used in this study are very young. Is there a reason to use 6-week old mice instead of older mice?
- There is no description of the behavioral assays conducted. Please expand and include details on the methodology of the Y-maze, novel object recognition, and Barnes maze tasks. Was a baseline behavior for each task conducted prior to the LPS administration? Were animals habituated to the apparatus for the behavior tasks prior to the LPS administration or at the start of the behavior following LPS and pharmacological agent administration? How many days were the animals tested? How was the behavior analyzed?
- Please include details on how many days following the pharmacological intervention that mice were collected for hippocampus and colon samples. Please also include details on how many days/hours following the behavioral tests that samples were collected.
- The behavioral test protocol could be included on the timeline of Figures 1 and 2 to add clarity.
- Based on Figure 2H, the intensity of confocal micrographs was analyzed. There is no detail on that analysis in the methods.
- Was there a computer program or statistical software that was used to complete the ANOVA tests and Duncan’s multiple range tests?
- Please thoroughly check figure legends for completeness and accuracy. In several of the figure legends, some of the panels are not included. For example, in Figure 1 legend there is no description of panel B.
- Thoroughly check that the figure references in the manuscript match the figure as well. The description in line 179 of Figure 2E-G doesn’t match what is described in the text with the panels of the figure.
- It seems like the authors overstate the combinatorial effects of donepezil with DW (DD group). In the figures, it doesn’t appear that the effect is significantly stronger following this treatment compared to the other DP, C29, DW, or DC groups. Are there statistical values that can be included to demonstrate this effect?
- Please make the case for the correlations more clear. What conclusions can be made by the correlations between the behaviors and gut bacteria populations or the inflammation in the hippocampus and the gut bacteria populations?
Author Response
We greatly appreciate your excellent suggestions. We revised our manuscript according to the suggestions of you and reviewers. The sentences in the manuscript revised by your comments are yellow-highlighted. The sentences revised for grammatical errors and repetition were not indicated. The revised manuscript was checked by a native speaker (www.supreme-trans.co.kr).
---------------------------------------------------------------------------------
The study of the microbiome-gut-brain relationship is in need of more exploration. Lee and colleagues explored this relationship using lipopolysaccharide (LPS)-induced cognitive impairments with donepezil and various combinations of probiotics. This study has high translational implications across multiple fields. In the manuscript titled “DW2009 elevates the effect of donepezil against cognitive impairment in mice,” the authors investigate the cognitive ability in mice following LPS-exposure along with the Alzheimer’s therapy of donepezil in combination with Lactobacillus plantarum C29 and DW2009 (a C29-fermented soybean).
To improve the manuscript for publication, below are a few items for consideration:
1. Throughout the manuscript, check for consistent and accurate grammar and spelling. For example, on line 50, Lactobacillus helveticus should be in italics, and on line 96, the ‘was’ should be a ‘were’, and on line 104, the first sentence is not a complete sentence. There were several other locations in the manuscript with other errors that impact the readability of the manuscript.
--> Thank you. We revised our manuscript including L50, L87, L95 overall. And the revised manuscript was checked by a native speaker (www.supreme-trans.co.kr).
2. The initial sentences of the introduction are scattered and could be connected better. Aging is discussed here, but not brought up again in the manuscript. This could be removed and the connection to inflammation and NF-kB signaling connected in the next sentence.
--> Thank you. We revised the first paragraph of Introduction section according to your comment (L30-L36).
3. The mice used in this study are very young. Is there a reason to use 6-week old mice instead of older mice?
--> Thank you. We hope to examine the effects of test agents on NF-kB activation-induced cognitive impairment. In the preliminary study, we can prepare mice with cognitive impairment by the intraperitoneal injection of LPS, as previously reported. Therefore, we also used young mice in the present study,
4. There is no description of the behavioral assays conducted. Please expand and include details on the methodology of the Y-maze, novel object recognition, and Barnes maze tasks. Was a baseline behavior for each task conducted prior to the LPS administration? Were animals habituated to the apparatus for the behavior tasks prior to the LPS administration or at the start of the behavior following LPS and pharmacological agent administration? How many days were the animals tested? How was the behavior analyzed?
-->Thank you. We revised the schedule in Figures 1 and 2. To measure cognitive behaviors to make a baseline, we measured cognitive behaviors in mazes. However, mice memorized the mazes. Therefore, we could not make a baseline. Therefore, to reduce the errors, the present studies were performed by skillful experts. The experimental protocols were described in L111-L130.
5. Please include details on how many days following the pharmacological intervention that mice were collected for hippocampus and colon samples. Please also include details on how many days/hours following the behavioral tests that samples were collected.
--> Thank you. We revised the schedule in Figures 1 and 2 and L105-L110.
6. The behavioral test protocol could be included on the timeline of Figures 1 and 2 to add clarity.
-->Thank you. We revised the schedule (a) in Figures 1 and 2.
7. Based on Figure 2H, the intensity of confocal micrographs was analyzed. There is no detail on that analysis in the methods.
--> Thank you. We added it in L151.
8. Was there a computer program or statistical software that was used to complete the ANOVA tests and Duncan’s multiple range tests?
--> Thank you. We added in L159-L160 and Supplementary material Table S1 and L161-162.
9. Please thoroughly check figure legends for completeness and accuracy. In several of the figure legends, some of the panels are not included. For example, in Figure 1 legend there is no description of panel B.
--> Thank you. We revised it.
10. Thoroughly check that the figure references in the manuscript match the figure as well. The description in line 179 of Figure 2E-G doesn’t match what is described in the text with the panels of the figure.
--> Thank you. We revised it (L200-L204).
11. It seems like the authors overstate the combinatorial effects of donepezil with DW (DD group). In the figures, it doesn’t appear that the effect is significantly stronger following this treatment compared to the other DP, C29, DW, or DC groups. Are there statistical values that can be included to demonstrate this effect?
--> Thank you. We added all F and p values in Supplementary material. And we revised the final conclusion according to your comment. (Line23-25, L345-L350, L384-385)
12. Please make the case for the correlations more clear. What conclusions can be made by the correlations between the behaviors and gut bacteria populations or the inflammation in the hippocampus and the gut bacteria populations?
--> Thank you. We revised our manuscript according to your suggestion (L370-377).
Reviewer 2 Report
I find the current study to be of interest to a wide variety of neuroscience researchers. The manuscript is generally well written, there are some english grammar corrections and awkward phrasing that need to be addressed. Memory behaviors is an awkward usage; cognitive tests, learning and memory, or cognitive behaviors would be a better usage. The behavioral tasks that were selected are appropriate; however, the descriptions of there usage is greatly lacking. I could never replicate their methods based on what is being reported. There are many different versions/variants of the y-maze, NOR and Barnes maze that makes interpretation of this data difficult. Number of days of testing, inter-trial intervals, habituation latencies all are important factors. In general the statistical analyses are fine; however, Barnes maze is using repeated days of testing, thus a repeated measures ANOVA is required. There is also no reporting of actual F-values, just astericks on a graph.
Author Response
We greatly appreciate your excellent suggestions. We revised our manuscript according to your comments. The sentences in the manuscript revised by your comments are yellow-highlighted. The sentences revised for grammatical errors and repetition were not indicated. The revised manuscript was checked by a native speaker (www.supreme-trans.co.kr).
-------------------------------------------------------------------------------------
I find the current study to be of interest to a wide variety of neuroscience researchers. The manuscript is generally well written, there are some english grammar corrections and awkward phrasing that need to be addressed.
--> Thank you. We revised many errors in our manuscript. And the revised manuscript was checked by a native speaker (www.supreme-trans.co.kr)
Memory behaviors is an awkward usage; cognitive tests, learning and memory, or cognitive behaviors would be a better usage. The behavioral tasks that were selected are appropriate; however, the descriptions of there usage is greatly lacking. I could never replicate their methods based on what is being reported. There are many different versions/variants of the y-maze, NOR and Barnes maze that makes interpretation of this data difficult.
--> Thank you. We added the behavioral tasks (L111 – L130) and the schedules (Figures 1 A and Figure 2A)
Number of days of testing, inter-trial intervals, habituation latencies all are important factors. In general the statistical analyses are fine; however, Barnes maze is using repeated days of testing, thus a repeated measures ANOVA is required. There is also no reporting of actual F-values, just astericks on a graph.
--> Thank you. We performed experiments more than duplicate. And we added its related F and P values in Supplementary material Table S1.
Round 2
Reviewer 1 Report
Thank you for the extensive revisions to the manuscript. The changes help with the clarity and impact of the study. One final consideration is to include the methods regarding confocal imaging and intensity analysis. How was this conducted? Were the settings on the confocal kept consistent? Was there a z-stack taken or just a single plane? How was the stack or plane selected? Was the analysis conducted on the 40x or 200x image? Was the intensity of the entire image analyzed or just specific regions of interest?
Additionally, a scale bar in Figure 2H would be helpful.
Author Response
Thank you for the extensive revisions to the manuscript. The changes help with the clarity and impact of the study. One final consideration is to include the methods regarding confocal imaging and intensity analysis. How was this conducted? Were the settings on the confocal kept consistent? Was there a z-stack taken or just a single plane? How was the stack or plane selected? Was the analysis conducted on the 40x or 200x image? Was the intensity of the entire image analyzed or just specific regions of interest?
-->Thank you for your excellent comment. We described it (L151-15 2, L221-L222, L245-L246)
Additionally, a scale bar in Figure 2H would be helpful.
-->Thank you. We revised Figure 2H and Figure 3F.